# A Layered Adopter-Structure Model for the Download of COVID-19 Contact Tracing Apps: A System Dynamics Study for mHealth Penetration

**DOI:** 10.3390/ijerph19074331

**Published:** 2022-04-04

**Authors:** Makoto Niwa, Yeongjoo Lim, Shintaro Sengoku, Kota Kodama

**Affiliations:** 1Graduate School of Technology Management, Ritsumeikan University, Ibaraki 567-8570, Japan; gr0486se@ed.ritsumei.ac.jp; 2Discovery Research Laboratories, Nippon Shinyaku Co., Ltd., Kyoto 601-8550, Japan; 3Faculty of Business Administration, Ritsumeikan University, Ibaraki 567-8570, Japan; lim40@fc.ritsumei.ac.jp; 4Life Style by Design Research Unit, Institute for Future Initiatives, University of Tokyo, Tokyo 113-0033, Japan; ssengoku@ifi.u-tokyo.ac.jp; 5Center for Research and Education on Drug Discovery, The Graduate School of Pharmaceutical Sciences, Hokkaido University, Sapporo 060-0812, Japan

**Keywords:** system dynamics, product growth model, COVID-19, contact tracing apps, notification apps

## Abstract

(1) Background: Contact tracing and notification apps for coronavirus disease 2019 (COVID-19) are installed on smartphones and are intended to detect contact with another person’s device. A high installation rate is important for these apps to enable them to be effective countermeasures against the silent transmission of diseases. However, the installation rate varies among apps and regions and the penetration dynamics of these applications are unclear. (2) Methods: The download behavior of contact tracing applications was investigated using publicly available datasets. The increase in downloads was modeled using a system dynamics model derived from the product growth model. (3) Results: The imitation effects present in the traditional product growth model were not observed in COVID-19 contact tracing apps. The system dynamics model, without the imitation effect, identified the downloads of the Australian COVIDSafe app. The system dynamics model, with a layered adopter, identified the downloads of the Japanese tracing app COCOA. The spread of COVID-19 and overall anti-COVID-19 government intervention measures in response to the spread of infection seemed to result in an increase in downloads. (4) Discussion: The suggested layered structure of users implied that individualized promotion for each layer was important. Addressing the issues among users who are skeptical about adoption is pertinent for optimal penetration of the apps.

## 1. Introduction

### 1.1. Challenges in Controlling the Spread of COVID-19

The coronavirus disease 2019 (COVID-19) pandemic was characterized by a large percentage of infected people with mild disease severity; therefore, the spread of infection associated with the behavior of mildly ill or asymptomatic individuals was remarkable [1]. The containment of the spread of COVID-19 by isolation of symptomatic individuals, which was effective in the case of new infectious diseases such as new strains of influenza, MERS, and SARS, was not as effective.

One possible measure to control the increase in such silent transmission is to conduct large-scale virus testing and isolate those who test positive [2]. However, a challenge with this approach is the feasibility of securing a large supply of virus testing kits as well as testing personnel and locations. In the case of countries with large land areas, large-scale virus testing in a short period is especially challenging because of the difficulty of distributing kits and the movement of people over long distances. Moreover, infection testing is problematic in low- and middle-income countries, owing to financial constraints and the lack of domestic capacity to manufacture nasal swabs or analytical reagent kits [3].

### 1.2. The Use of mHealth (Contact Tracing Apps) in Controlling COVID-19

Communication technology seems to be an effective medium in addressing problems concerning a vast land area. Although remote diagnosis [4] is one solution, procuring human resources can become a limiting factor. Therefore, it is desirable that the measures for controlling the spread of the COVID-19 infection be more labor saving [5]. From this perspective, the idea that effective isolation could be achieved by conducting contact tracing was developed after finding that the COVID-19 infection is mainly caused by close contact. The introduction of contact tracing or contact notification applications (apps) was an attempt to achieve this goal. A typical app is installed on a smartphone and uses Bluetooth communication to detect contact with another person’s device. Although various ways of utilizing this app-based data were being explored, the primary goal was to control the spread of infection by identifying and isolating individuals who had come into contact with an infected person.

COVID-19 tracing apps were a novel form of mobile health (mHealth) technology. mHealth is both a product and service that applies information and communication technologies, including mobile information devices, to the field of healthcare and is widely used for disease prevention and health promotion [6]. Many studies have been previously conducted to assess the penetration of mHealth technologies [7] and electronic health records (EHRs) [8], which are a form of eHealth. Based on this premise, the present study focused on examining the penetration of COVID-19 tracing apps.

The appropriate use of mHealth, COVID-19 contact tracing apps, and health promotion apps during the pandemic could be of interest from the viewpoint of maintaining people’s health during a pandemic. Health promotion was considered extremely important during the pandemic [9], and this study aimed to provide information on the penetration of mHealth during the COVID-19 outbreak. In addition, mHealth is being actively developed by pharmaceutical companies [10] to provide comprehensive healthcare services. Thus, the penetration dynamics of mHealth are important for pharmaceutical companies for their potential development candidates for mHealth. As such, the present study examined how the penetration model, as a case study of mHealth against infectious diseases, could contribute to health promotion in society, including the business sustainability of pharmaceutical companies.

For contact tracing apps to be effective, they need to communicate with the applications possessed by the contacts post-installation. There are two positions on the installation rate: one is to assume that a certain number of installations (typically 60% or more) is necessary to contain the infection through countermeasures depending on the apps alone [11], and the other is to expect that a lower installation rate (not as high as 60%) still has a significant effect on infection control [12]. In either case, a higher installation rate is preferable. Therefore, in countries that adopted these approaches, governments were encouraging citizens to install the relevant software. As a result, some countries witnessed extensive downloads in a short period, while others did not see much progress. For example, the Australian app COVIDSafe was downloaded by more than 20% of the total population [13] in two weeks [14], while the French app StopCOVID (now called TousAntiCovid) plateaued at 2% [15]. The number of downloads per day for apps was seen to be generally high for a week or two after introduction, after which it then decreased. The cumulative number of downloads varied depending on the situation in each country. Thus, the degree of penetration seemed to vary depending on the environment. Therefore, it is important to analyze the background and structure of contact tracing apps’ penetration and explore appropriate strategies to promote penetration.

### 1.3. Purpose of the Study

In the present study, we analyzed the factors that affected the download rate of COVID-19 contact tracing apps using a system dynamics approach to examine a feasible strategy against new infectious diseases with similar characteristics in the future. Simultaneously, this analysis of COVID-19 contact tracing apps was a suitable case study on mHealth penetration, and, therefore, made a valuable academic contribution to the research on mHealth. Furthermore, this study aimed to assist pharmaceutical companies in promoting the adoption of mHealth, especially when commercializing it as a countermeasure for infectious diseases that spread over a short period.

### 1.4. Theoretical Framework and Research Questions

#### 1.4.1. New Product Growth Model as a Base Model and (Non-)Conformity Testing

Previous studies on mHealth penetration include Hwang’s research on mHealth for diabetes management [7]. Hwang stated that the new product growth model for consumer durables by Bass et al. [16] applied to the sphere of mHealth. Otto et al. stated that the same model applied to the introduction of EHRs as well [8]. This product growth model developed by Bass et al. stated that the speed of new product introduction (the number of sales per unit of time, which Bass defined as the purchase probability f(T) at time T) depended on the fixed effect of the innovator (a consumer who actively introduces a new product, related to the innovation coefficient p) and the variable effect of the imitator (a consumer who introduces a product by imitating the behavior of others, related to the imitation coefficient q and the product penetration rate). Expressing this model in a mathematical equation, the purchase probability of a new product at time T, f(T), is expressed as f(T) = [p + q F(T)][1 − F(T)], where F(T) is the penetration rate up to that point. In other words, immediately after the initial introduction of a new product, the introduction rate depends on penetration (an increase in F(T)) and increases owing to the increase in qF(T). When the product is introduced beyond a certain level, the introduction rate decreases owing to the decrease in [1 − F(T)].

Originally, the product growth model developed by Bass et al. was related to product growth in a free market. A preceding study by Hwang et al. also examined the penetration of profitable mHealth for diabetes in the free market. On the other hand, the study by Otto et al. stated that Bass’s model could be applied to the penetration of EHRs, which are not regarded as consumables, using the model to describe the effect of awareness of the product. COVID-19 tracing apps are usually free of charge and are distributed by government initiatives. In addition, the distribution of COVID-19 tracing apps was aimed not only at individual users but also at the entire society as a measure to control the spread of disease. Based on this understanding, Bass’s product growth model was employed in the present study as an initial model to analyze the essential components of product adoption by users. Initially, the effect of awareness among users was included as an imitation effect.

First, the (non-)conformity of the product growth model was investigated. We examined whether the penetration of the COVID-19 tracing app conformed to the model in order to obtain insights into the penetration dynamics. The hypotheses are presented below as research question (RQ) RQ1 and hypothesis (H) H1.

**RQ1.** 
*Can Bass’s new product growth model explain the download behavior of contact tracing apps during a pandemic?*


**H1.** 
*The penetration dynamics of the introduction of COVID-19 tracing apps can be explained through Bass’s new product growth model.*


#### 1.4.2. Implementation Behavior of COVID-19 Contact Tracing Apps

Because a COVID-19 contact tracing app uses some form of behavioral information, privacy concerns were raised [11]. In addition, due to the nature of tracing and preventive quarantine wherein people are isolated based on contact with virus carriers regardless of whether they are infected or not, the direct benefit to the person who installed the app is unclear. Although the entire society benefits from the suppression of the spread of infection, being quarantined without being infected can be annoying for people, mainly due to the loss of time.

The above disadvantages may be factors preventing the installation from proceeding as expected or desired. Prior research on this perspective includes a study by Garrett et al. [17], who conducted an online survey to identify the level of acceptance of several tracing technologies in Australia before the introduction of the COVIDSafe app. An approach such as that of the COVIDSafe app, in which the introduction of the app was voluntary and the infected person enters his or her information voluntarily, the ease of physical distancing based on the use of the app is maintained and is generally well accepted, as shown in the study.

The major reasons for not downloading COVIDSafe were found to be issues related to data security (e.g., preserving privacy, normalizing government tracing, and a belief that the government is not trustworthy) and functionality (e.g., battery drainage and a belief that the app is not effective). Some of the reasons for not installing the app were based on the lack of knowledge of the COVIDSafe tracing technology. Another reason for not installing the app was that some people wanted to wait for others to install it first. The major motivations for installation were to ensure one’s own health and to be able to live a normal life without behavioral restrictions.

Chan et al. [18] found that resistance towards downloading and using the contact tracing app was due to concerns about violating personal privacy. In other words, as with EHR, adoption behavior was determined by the adopter’s decision as to whether the benefit to the public (and the benefit to the individuals included in the public) or the detriment to the individual was more important [19]. Studies have examined the relationship between salient disease concerns and intentions to download in France, salient disease concerns and social conservatism in Australia, and privacy concerns in the United States. Overall, the results indicated that the pandemic reduced the intention to download contact tracing apps due to enhanced privacy concerns. This finding indicated that pandemics make society more conservative and sensitive to individual detriments.

Based on previous studies, the second research question of the present study was to determine the influence of the spread of infection as a factor that may affect the penetration of the application (RQ2). We formulated the hypothesis H2a “installation decreases due to pandemics”, which directly corroborated the results of Chan et al.’s cross-sectional study [18]. On the other hand, as concerns about one’s own health [17] motivated installation, another hypothesis H2b “installation may be promoted due to increased concerns about one’s own infection caused by the spread of infection” was formulated.

**RQ2.** 
*Is the downloading behavior of contact tracing apps during a pandemic reduced or facilitated by infection concerns? Alternatively, there may be other influencing factors.*


**H2a.** 
*Spread of disease decreases downloads via enhanced privacy concerns.*


**H2b.** 
*Spread of disease increases downloads via people’s concerns about their own health.*


Another hypothesis was developed based on the fact that the COVIDSafe app was positioned as a means to return to normal life and even venture out of homes, which was said to be the incentive for the download [14]. Hypothesis H2c was based on the idea that an enhanced intention to move out and around, triggered by relaxation of behavioral restrictions, might increase the intention to download and install contact tracing apps.

**H2c.** 
*Relaxation of behavioral restriction increases installations.*


#### 1.4.3. Layered Structure of Potential Users

A summary of previous studies on COVID-19 tracing apps indicated that potential users had a layered structure, classified by the perceived health risks of infection. Blom et al. [20] conducted a web-based survey on the factors that influenced the adoption of the Corona-Warn-App in Germany. The results denoted that 81% of people in the age group of 18–77 years had the ability to install the app, but only 35% were willing to do so. It was also found that those at risk of serious illness or death from an infection were more willing to install the app than those at a lower risk. This willingness, possibly related to the personal perceived risk of infection, might have been related to the early adoption trend of contact tracing apps.

Garrett et al. [17] reported that confirmed COVIDSafe downloads were far fewer than those predicted by survey-based acceptability. This implied that survey-based acceptability might not translate into public uptake. This finding suggested that survey-based grouping could be divided into subgroups in real-time settings because of the multiple factors influencing human behavior (as highlighted in the previous section).

On the other hand, even for a quick and successfully penetrated app such as COVIDSafe, installation plateaued at 20% of the total population [14]. This observation implied that only a portion of users responded to the initial installation. Based on previous studies and observations, we formulated the hypothesis that potential users have a layered structure.

**RQ3.** 
*Can the downloading behavior of contact tracing apps be described as a layered user structure?*


**H3.** 
*Potential users have a layered structure classified by perceived health risks associated with infection.*


### 1.5. Significance of the Study

Adoption of new technologies under pandemic conditions are attracting interest. These typically include information systems [21,22] enabling physical distancing. One study investigated factors related to choosing technologies [23] based on planned behavior theory using a qualitative approach. The current study used a modeling technique and quantitative data to approach technology adoption from a different angle. The advantage of this approach was the ability to use quantitative time-course data to gain insights into adoption dynamics and the structure behind those dynamics.

This study was unique in that it examined mHealth penetration in a short period during a pandemic. As previously mentioned, Hwang [7] studied mHealth penetration using system dynamics, which examined economic value over a long period, and EHR was another technological approach to communication over distance [8], but penetration in the context of a pandemic has not yet been studied. In this study, we focused on contact tracing apps, whose penetration could contribute to public health. We further examined the structure behind that penetration.

## 2. Materials and Methods

### 2.1. Studied Apps

In this study, we focused on government-sponsored anti-COVID-19 apps that were widely adopted in various countries and which had the following features: users voluntarily provided information, the information provided was not used for any other purpose, and users’ information was anonymized or not referenced [24]. Apps with this feature were widely in use at the time the study was conducted. Further, for the quantitative study, we mainly studied COVIDSafe (Australia) and COCOA (Japan), which had detailed installation numbers systematically reported by each government [14,25]. They were two apps among twenty moderately to highly rated contact tracing apps used worldwide, summarized by Chaudhary et al. [26].

Both COVIDSafe and COCOA had contact detection ability via Bluetooth technology; however, they differed in the details of their tracing technology. COVIDSafe used a centralized processing protocol [17], while COCOA used decentralized data handling. COVIDSafe was employed as an example of quickly penetrated apps, despite potential privacy concerns. COCOA was regarded as an example of slow and stepwise penetration.

### 2.2. Modeling Approach

#### 2.2.1. General Principle

The modeling was conducted via exploratory semi-quantitative analysis describing the factors related to the adoption of contact tracing apps, based on theories from previous studies. The semi-quantitative model was represented as a stock and flow diagram of the change in the number of users at each stage of adoption. The interrelationships between the factors in the model were displayed by connecting them with arrows. The relationships between factors were indicated by positive (+) arrows, when they were mutually reinforcing, and negative (−) arrows, when they affected each other in the opposite direction. This relationship was established based on previous studies (existing in the literature). In addition, both hypothetical and empirical relationships could be assumed in the system dynamics practice [27]. The figures in the text refer to the conceptual model, while the actual calculations were based on the stock and flow model created using Vensim Professional software.

First, the model was constructed based on the mind model concept in system dynamics practice considering Bass’s product growth model (Figure 1a) to explain real-world functions in a semi-quantitative manner, and, then, the goodness of fit was examined visually. The accuracy of the model was verified by examining the model equations and units of variables. The validity of the model was verified by inspecting each model to determine whether the necessary variables and connections between variables were expressed using a white-box approach [28]. The validity of the model structure was tentatively considered acceptable when real-world events could be described by parameter optimization. If the events could not be represented, even after optimization, the model structure was considered invalid (Figure 1b).

#### 2.2.2. Contact Tracing App Downloads and New Product Growth Model

First, based on previous research, a quantitative stock-flow model was developed based on Bass’s new product growth model with the addition of influencing factors, which indicated the number of downloads as a stock. Equations between stocks followed the basic concepts of Bass’s product growth model, as described in Section 1.4.1. The dynamics of the model were then examined while changing the parameters.

Next, quantitative data on the transition in the number of downloads were obtained from the government’s public information system and compared with the behavior of the model. In particular, it was examined whether a rapid increase in the number of downloads in the initial phase and subsequent transition to the low-growth phase, which is characteristic of the new product growth model, could be observed in the present analysis.

#### 2.2.3. Contact Tracing App Downloads and Concern on Disease Spreading

Based on the Australian COVIDSafe model, which took a short time to download, we created a Japanese COCOA app model, which took a long time to penetrate. As system dynamics require consideration of the model structure when creating the model, we began by examining the actual COCOA app data and considered ways to extend the COVIDSafe model structure to make it closer to the actual data. In the process of creating the model, we examined how the spread of disease affected downloads. We confirmed that the effect observed in the COCOA app was also observed in the Corona-Warn-App in Germany.

Based on the observations, a system dynamics model of the COCOA app was created.

## 3. Results

### 3.1. (Non-)Conformance to Bass’s New Product Growth Model (COVIDSafe as a Case Study)

First, a quantitative stock-flow model was created based on Bass’s new product growth model (Figure 2), and the ratio of the innovation coefficient to the imitation coefficient was set based on Song’s previous study. Detailed modeling information is provided as Appendix A. Under this condition, where the elapsed time was arbitrary, the number of downloads took the form of a sigmoidal curve (Figure 3). In other words, because download readiness via the imitation effect was influenced by downloaded users, downloads increased based on positive feedback. When the number of downloads exceeded a certain level, saturation was observed owing to a decrease in potential users. Consequently, the number of downloads indicated a sigmoidal growth curve. However, in real-world observations, the COVIDSafe and COCOA apps indicated a rapid increase followed by a plateau (Figure 4) and did not exhibit as a sigmoidal growth curve. This might be because the COVID-19 contact tracing app download examined in this study was observed for a short period with a focus on the innovator’s behavior. According to Sterman [29], the acceleration of adoption was due to word-of-mouth through contact with the adopter, and the imitation effect might be less pronounced under conditions where human interaction was restricted.

In Bass’s new product growth model, a plateau was reached when the product was distributed; however, in the real-world case, the plateau was reached before the product was distributed to those with access to it (81% according to Blom et al. [20]). A plateau might be reached once the product had been distributed to those with the intent to install it. According to Blom, 38% of people had the intention to install the product, and if 38% of the 81% of people with access intended to install the product, it was estimated that 31% had the intention to install the product.

Based on the aforementioned considerations, we developed an improved model to explain the download behavior of the COVIDSafe app (Figure 5). In the improved model, the biphasic nature of the innovator and imitator was eliminated, and the model was monophasic with only the innovator. The improved model accurately indicated the download behavior of the COVIDSafe app (Figure 6).

As described above, the case study of the COVIDSafe app showed that the penetration of contact tracing apps during a pandemic might not indicate the imitation effect similar to Bass’s new product growth model. This might be because the imitator might be an uninstalled (un-downloaded) user in this observation period, or the imitation effect (i.e., word of mouth communication) might not be significant under pandemic measures where human interaction is restricted. In addition, the short-term potential adopters were estimated to be about 25%, which was a significant discrepancy from the 81% of accessible people estimated in Germany and was close to the number of people with installation intentions among accessible people.

### 3.2. Download Behavior of Contact Tracing Apps during a Pandemic (COCOA as a Case Study)

The data on the COCOA downloads indicated that they remained low in the short term. Although the downloads showed an increasing trend over a long period, they were characterized by the presence of two phases of downloads (Figure 7). Interventions that were characteristic of the second phase were retrieved from the Oxford COVID-19 Government Response Tracker [30], and other events were retrieved from various news sources. The number of positive cases was retrieved from the Oxford COVID-19 Government Response Tracker. The second phase (30–60 days after release, July–August 2020) corresponded to the period of re-emergence of the disease when government intervention was slightly intensified (Figure 7). On the other hand, downloads responded slowly to disease spread after day 150. As for other events, the period around day 30 also corresponded to the implementation of the “Go to Travel Campaign”, a promotional program initiated by the Japanese government to promote the travel and hotel industry (from 22 July 2020, to 33 days after the introduction of the app).

The current case study suggested that the spread of infection and the expansion of the movement area led to an increase in the number of downloads.

### 3.3. Layered Structure of Potential Users (COCOA as a Case Study)

Based on the previous discussion, we developed a model for the COCOA app where penetration was slow and there were influencing factors, such as the spread of disease and extending the range of movement, which were different from the case where penetration was achieved in a short period, similar to COVIDSafe.

Initially, following Bass’s model of new product penetration of homogeneous potential users, the initial adopters were set as innovators and additional adopters, because of influencing factors, including the expansion of behavioral range, were set as imitators (as in Figure 2). However, the saturation observed in the COCOA app within one month of introduction (Figure 7) was not expressed appropriately. By classifying potential users into layers, the saturation and gradual expansion of the download transition could be generally expressed (Figure 8 and Figure 9) in comparison with the single-layered model (Appendix A). In other words, it was suggested that the expansion of downloads due to influencing factors was not a result of imitation among homogeneous users but because of the expansion of layers of users that were stratified based on the adoption attitude of apps.

### 3.4. Causal Loop Diagram for the Understanding of Interrelationships among Items

Based on the above considerations, findings from preceding and current studies were organized into a causal loop diagram to emphasize the interrelationships among items (Figure 9). The observation that an increase in downloads did not follow a seamless sigmoid curve conformed with Bass’s growth model. This might be due to not only the short observation period but also the lack of word-of-mouth, as shown in Figure 10. On the other hand, saturation was observed in each user layer.

## 4. Discussion

The present study conducted a detailed analysis of the download behavior of COVID-19 contact tracing apps to identify countermeasures against new infectious diseases in the future. The study employed Bass’s new product growth model using the COVIDSafe app in Australia as a case study. The analysis indicated that the imitation effect was not observed in the time range of one to two months during the pandemic, and Bass’s new product growth model did not work properly. Essentially, it was difficult to introduce new infection control apps using conventional penetration strategies.

A distinguishing aspect of this study was the short observation period and observation under pandemic conditions with physical distancing restrictions in place, which was different from those in which Bass’s product growth model [7] was used. Thus, the reason for non-conformance to Bass’s model could be due to the length of the observed period (Bass’s model typically evaluated annual growth), or the lack of person-to-person contact (word-of-mouth communication related to products) due to physical distancing under pandemic conditions. Therefore, improving electronic/remote communication under restricted mobility could lead to an increase in penetration efficiency. Another reason for non-conformance was that examining the nature of the underlying behavior of people (Bass’s model is for products sold in the free market) was possible. This specific feature also suggested that the factors that specified the adaptor layer could be related to people’s behavior in pandemic conditions. Several studies in Australia have suggested that some people do not install such apps immediately because they wait for others to install them [17,31]. It was also suggested that such users were uninstallers (non-downloaders). In addition, a study in Germany [20] suggested that those at a higher risk of serious illness or death after infection were more willing to install the apps than those at a lower risk. Overall, it was suggested that the original behavior of people was at least two-tiered, and people with high-risk health concerns were likely to download and install the app swiftly. Those waiting for others to install were not considered to download or install in the early phase after the app’s release.

We further investigated the download behavior of contact tracing apps during the pandemic using the COCOA app as a case study. The results indicated that the download increase might be related to the strengthening of government intervention measures in response to the spread of infection and influenced by the expansion of the scope of activities. Even in the case of the COCOA app, the diphasic relationship between the innovator and imitator was not clear. Observed up to nine months after the introduction of the app, the download increase was observed to be due to the addition of a new layer of users stratified by the adoption attitude of the application.

While examining the relationship between behavioral restriction relaxation and downloading, for the COVIDSafe app, which achieved a large number of downloads in a short period, the government took an initiative to promote the use of the app to ease behavioral restrictions from the time of release [25]. In the case of the COCOA app, the timing of the “Go to Travel Campaign” coincided with the time when the number of downloads increased again. From these observations, it was plausible that there was a relationship between the adoption of contact tracing apps and the relaxation of behavioral restrictions. In addition, the multifaceted activities to raise awareness of this kind of app might have played a role in the increase in downloads. With regard to the COCOA app, based on the appeal of the Japanese government, the Japan Business Federation requested that each company should raise awareness of infection prevention measures and that their employees should install the COCOA app as a measure for preventing the spread of infection in the workplace since December 2020 (six months after the release of the COCOA app) [32]. Universities in Japan were also raising awareness on measures to prevent the spread of infection and recommending the installation of the COCOA app [33]. Thus, the COCOA app might have gradually diffused owing to the success of these multifaceted measures.

Although the present study revealed important findings, it had several limitations. One of the limitations was that the study focused only on the number of downloads for which data were readily available, and thus provided limited insight into whether the application was being used. The tendency for apps to be uninstalled without being used for implementation was identified as a problem, and further research is needed on this topic. A previous study found that the level of trust in the government influenced the intention to implement [34]. In this study, the effects of trust in the government or unintended disturbances, such as initial malfunctions or potential security vulnerabilities, were not addressed and further research is required.

The findings from this study indicated that it was not advisable to expect an imitation effect of the countermeasures for new infectious disease control. A non-conformance to Bass’s product growth model highlighted the inadequacy of employing a penetration strategy similar to consumable durables in contact tracing apps. The suggested layered structure of users implied that individualized promotion for each layer was important. It might be useful to address the perception of problems among users who are skeptical of adoption. One potential approach is to provide incentives that exceed the importance of the perceived problems associated with contact tracing apps.

## 5. Conclusions

The present study demonstrated the characteristics of the penetration of new technology measures in a pandemic situation under physical distancing norms. The findings obtained from this study might be beneficial for developing strategies for the optimal penetration of mHealth under pandemic situations within a short period. Based on the findings obtained in this study, it would be beneficial for future studies to examine how the different relaxation styles of mobility restrictions can affect the penetration of anti-infectious mobile technologies.

## Figures and Tables

**Figure 1 ijerph-19-04331-f001:**
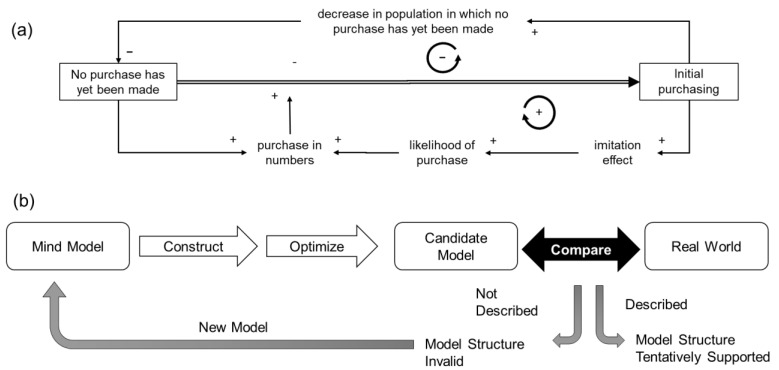
(**a**) Representation of Bass’s new product growth model by a system dynamics concept. (**b**) Conceptual schema of the approach employed in the current study.

**Figure 2 ijerph-19-04331-f002:**
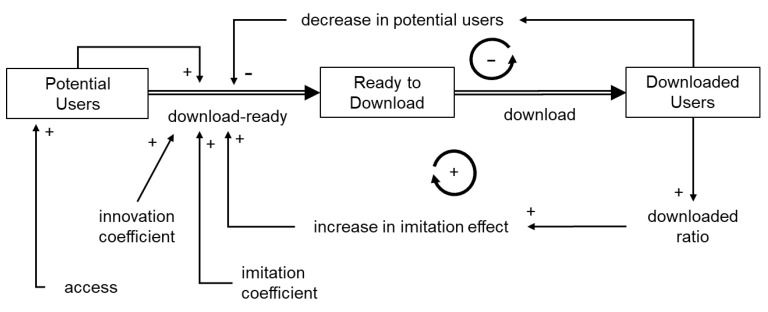
A conceptual system dynamics model of app downloads based on Bass’ new product growth model.

**Figure 3 ijerph-19-04331-f003:**
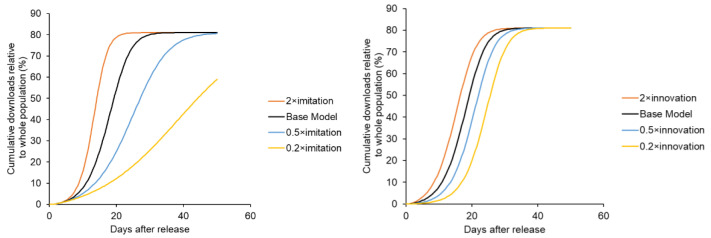
Growth curve of downloads based on Bass’s new product growth model with varying imitation factor (**left** panel) and varying innovation factor (**right** panel).

**Figure 4 ijerph-19-04331-f004:**
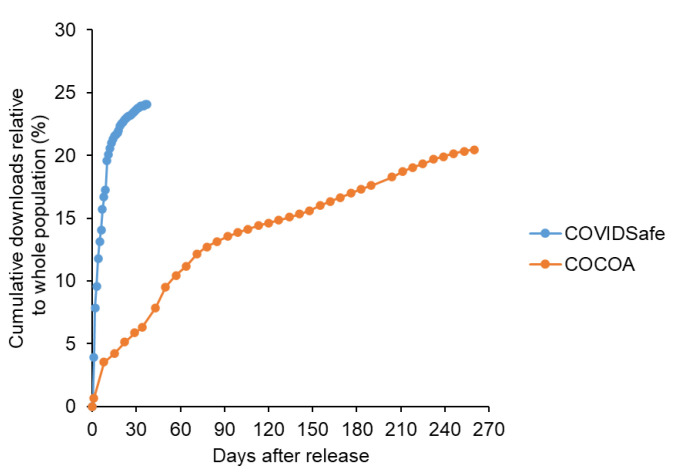
Confirmed downloads of COVIDSafe and COCOA apps after release.

**Figure 5 ijerph-19-04331-f005:**
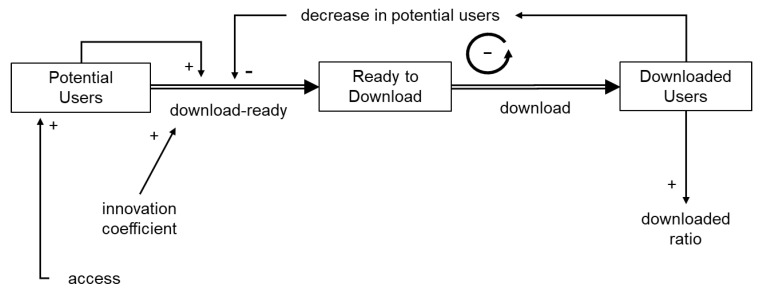
COVIDSafe’s conceptual system dynamics model on the number of downloads.

**Figure 6 ijerph-19-04331-f006:**
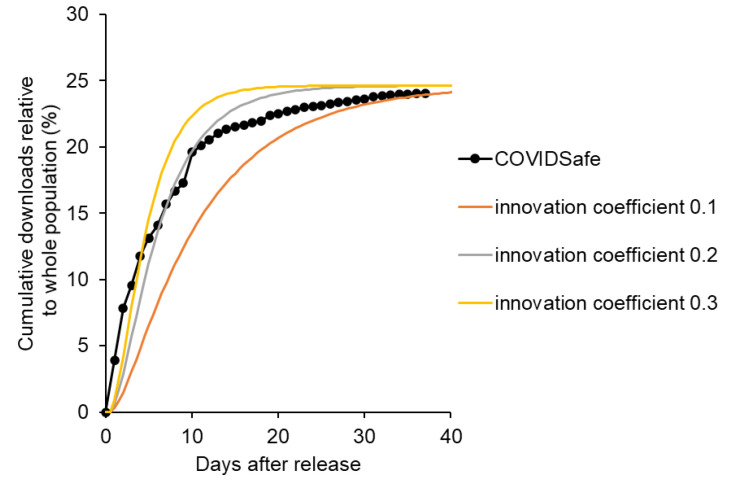
Representation of the number of downloads of COVIDSafe app (system dynamics model).

**Figure 7 ijerph-19-04331-f007:**
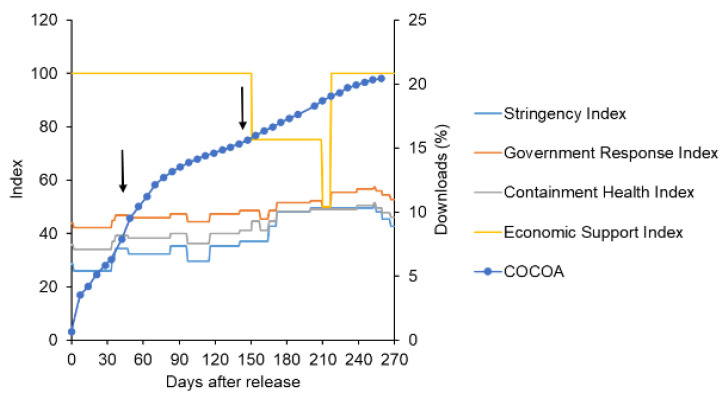
Relationship between the number of COCOA downloads and government interventions (upper panel) and the number of positive cases (lower panel).

**Figure 8 ijerph-19-04331-f008:**
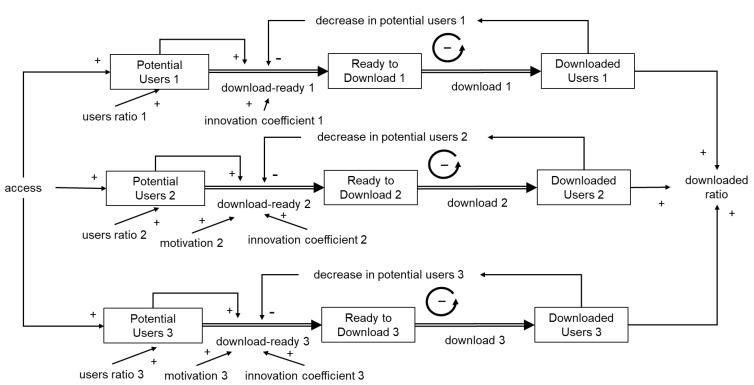
Conceptual system dynamics model of the number of COCOA downloads.

**Figure 9 ijerph-19-04331-f009:**
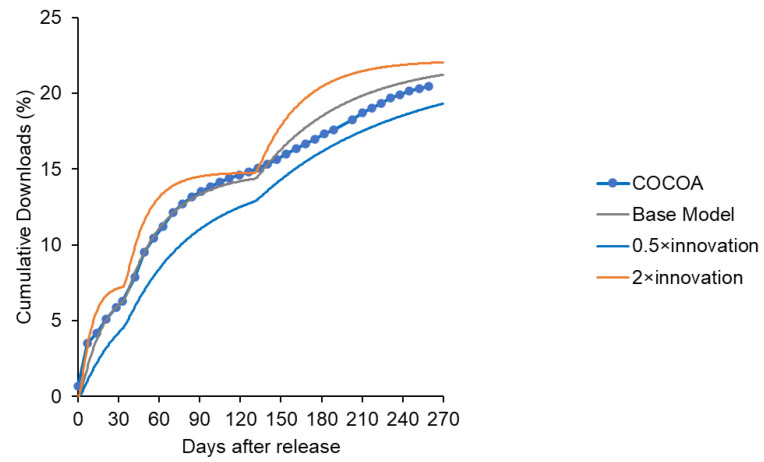
Representation of the change in the number of downloads of COCOA expressed by a system dynamics model with a layered structure.

**Figure 10 ijerph-19-04331-f010:**
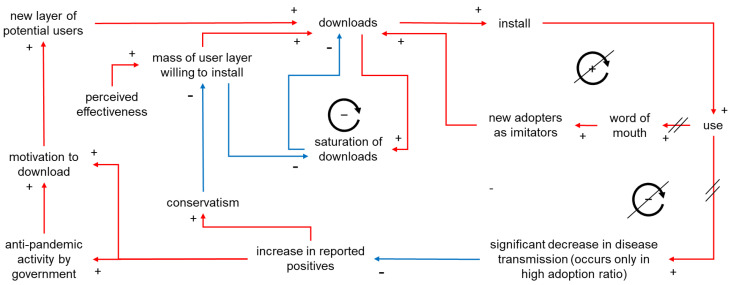
Causal loop diagram for the introduction of anti-COVID-19 contact tracing apps. Diagonal strike-out lines indicate that the effect was not observed in real-world settings. The circle with (+) indicates a reinforcing loop and circles with (−) indicate balancing loops. Red arrows indicate direct relationships and blue arrows indicate inverse relationships.

## Data Availability

Publicly available datasets were analyzed in this study. The government response to COVID-19 was obtained from Oxford COVID-19 government response tracker. Available online: https://covidtracker.bsg.ox.ac.uk/ (accessed on 15 January 2022). COVIDSafe downloads were obtained from the article “How many people have downloaded the COVIDSafe app and how central has it been to Australia’s coronavirus response?”, which is available online: https://www.abc.net.au/news/2020-06-02/coronavirus-covid19-covidsafe-app-how-many-downloads-greg-hunt/12295130 (accessed on 15 January 2022). Download number of COCOA was manually collected from the website of the Ministry of Health and Welfare: https://www.mhlw.go.jp/stf/seisakunitsuite/bunya/cocoa_00138.html (In Japanese) (accessed on 15 January 2022).

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
