# Peer review of "A Layered Adopter-Structure Model for the Download of COVID-19 Contact Tracing Apps: A System Dynamics Study for mHealth Penetration"

_ijerph, 2022, doi:10.3390/ijerph19074331_

Round 1

Reviewer 1 Report

The article presents an interesting approach to modeling behaviours with respect downloading apps. Authors arte based on existing models and make use of, in my opinion, scarce data. Hence, a study considering a longer period of time could be more interesting. However, due to the urgency and significance related to the covid topic, conducting larger studies is not straightforward. 

The article adresses covid tracing apps from the perspective of the “popularity” or “downloads” of the app. This approach is kind of original, in my opinion. I’m not confident about “marketing” or “economics”, but from the technology perspective, it is interesting to analyse these topics for an app, specially with the concerns (privacy) entailed to covid 19 pandemics. The paper is easy to understand, and addresses all the required sections in an appropriate manner. I am not expert in the market study areas, so I do not feel confident suggesting how to improve the paper from this point of view. 

Author Response

March 31, 2022

Dear Reviewer,

First of all, we’d like to express to you our gratitude. Your comments and suggestions are invaluable for us to improve our paper.

We believe that we did our best in clarification and improvements according to all of your comments and suggestions as below, and we hope that our responses meet your expectations and your intentions.

We highly appreciate your cooperation.

Warm Regards,

Kota Kodama, PhD

Graduate School of Technology Management, Ritsumeikan University

2-150, Iwakura-cho, Ibaraki, Osaka, 567-8570, Japan

+81-72-665-2448

[email protected]

ID

Comments and Suggestions

Response

1-1

The article presents an interesting approach to modeling behaviours with respect downloading apps. Authors arte based on existing models and make use of, in my opinion, scarce data. Hence, a study considering a longer period of time could be more interesting. However, due to the urgency and significance related to the covid topic, conducting larger studies is not straightforward.

Thank you for this comment. We agree to these viewpoints. Observation thorough longer period of time seems interesting and this will be challenge for future studies. Considering changing condition of covid issue, current study aimed to seek the best use of short-term data.

1-2

The article adresses covid tracing apps from the perspective of the “popularity” or “downloads” of the app. This approach is kind of original, in my opinion. I’m not confident about “marketing” or “economics”, but from the technology perspective, it is interesting to analyse these topics for an app, specially with the concerns (privacy) entailed to covid 19 pandemics. The paper is easy to understand, and addresses all the required sections in an appropriate manner. I am not expert in the market study areas, so I do not feel confident suggesting how to improve the paper from this point of view.

Thank you for this comment. We agree to these viewpoints and indeed this paper aimed to get insight in technology adoption using marketing model with a recognition of limitations.

Reviewer 2 Report

A system dynamics study for m-Health 3 penetration was presented by the authors. It is a model based on a layered adopter structure model for the download of COVID-2 19 contact tracing apps.

In their initiative, the authors' analysis of COVID-19 contact tracing apps is a suitable case study on mHealth penetration. Hence, It makes a valuable academic contribution to the research on mHealth in order to assist pharmaceutical companies in promoting the adoption of mHealth, especially when commercializing it as a countermeasure for infectious diseases that spread over a short period.

A State of the Art section is missing, as well as to introduce a comparative table in order to know the advantages and drawbacks in comparison with other similar proposals. Also, authors should include related works with high-quality references, such as indexed journals.

In the Materials and Methods section, the description of the Modeling Approach is very concise. However, I suggest including a graphical representation of it. I. e. is necessary to include a conceptual schema of the approach presented.

In addition to a conceptual schema, is necessary to include a brief features description of the COVIDSafe App and COCOA app. And, if is possible, add relevant differences between them and specify, Why did the authors select these Apps? This is for the purpose of improving and supporting the provided information provided in this section.

The conclusions section is not included in the paper. Additionally, I would like to suggest in this section, future works due to the relevance of this work

Author Response

March 31, 2022

Dear Reviewer,

First of all, we’d like to express to you our gratitude. Your comments and suggestions are invaluable for us to improve our paper.

We believe that we did our best in clarification and improvements according to all of your comments and suggestions as below, and we hope that our responses meet your expectations and your intentions.

We highly appreciate your cooperation.

Warm Regards,

Kota Kodama, PhD

Graduate School of Technology Management, Ritsumeikan University

2-150, Iwakura-cho, Ibaraki, Osaka, 567-8570, Japan

+81-72-665-2448

[email protected]

ID

Comments and Suggestions

Response

2-1

A State of the Art section is missing, as well as to introduce a comparative table in order to know the advantages and drawbacks in comparison with other similar proposals. Also, authors should include related works with high-quality references, such as indexed journals.

Thank you for this comment. At the end of introduction section, “significance of the study” subsection is built and advances of the current work, not only from mHealth but also general technology adoption perspective is described in relation to related studies, adding 3 more references from indexed journals. Importance of studied apps are added in Methods section in response to another comment (ID 2-3).

2-2

In the Materials and Methods section, the description of the Modeling Approach is very concise. However, I suggest including a graphical representation of it. I. e. is necessary to include a conceptual schema of the approach presented.

Thank you for this comment. New Figure 1 was added to describe these items. Figure 1(a) describes graphical representation of Bass’s product growth model and Figure 1(b) describes conceptual schema of approach employed in the study

2-3

In addition to a conceptual schema, is necessary to include a brief features description of the COVIDSafe App and COCOA app.

Thank you for this comment. New section 2.1 was made and studied apps are described.

2-4

And, if is possible, add relevant differences between them and specify, Why did the authors select these Apps? This is for the purpose of improving and supporting the provided information provided in this section.

Thank you for this comment. In new section 2.1, the reason for the selection of the apps (COVIDSafe: quick penetration, COCOA: slow and stepwise penetration) is described.

2-5

The conclusions section is not included in the paper. Additionally, I would like to suggest in this section, future works due to the relevance of this work.

Thank you for this comment. Conclusion section is built, and potential future work is described in it.

Reviewer 3 Report

The paper is interesting and the results are significant to model how COVID Apps were adopted by the users.

 Nevertheless, the methods and models used to evaluate the adaption rate (that is. section 3) are not clearly described. Although you include some schemes (figures 1,4 and 7), I don't see how these conception models are finally implemented using Bass's model.

Therefore, it is required to detail these models.

MINOR ISSUES:

In the following text you haven't defined NPBM:

RQ1: Can Bass’s NPGM explain the download behavior of contact tracing apps dur- 149 ing a pandemic?

Author Response

March 31, 2022

Dear Reviewer,

First of all, we’d like to express to you our gratitude. Your comments and suggestions are invaluable for us to improve our paper.

We believe that we did our best in clarification and improvements according to all of your comments and suggestions as below, and we hope that our responses meet your expectations and your intentions.

We highly appreciate your cooperation.

Warm Regards,

Kota Kodama, PhD

Graduate School of Technology Management, Ritsumeikan University

2-150, Iwakura-cho, Ibaraki, Osaka, 567-8570, Japan

+81-72-665-2448

[email protected]

ID

Comments and Suggestions

Response

3-1

The methods and models used to evaluate the adaption rate (that is. section 3) are not clearly described. Although you include some schemes (figures 1,4 and 7), I don't see how these conception models are finally implemented using Bass's model. Therefore, it is required to detail these models.

Thank you for this comment. Bass’s model was clearly presented as new Figure 1 (a) for the readers to compare the structure of new Figure 2. In addition, comments stating that model details are described in supplementary, was added in section 3.1.

3-2

In the following text you haven't defined NPGM:

RQ1: Can Bass’s NPGM explain the download behavior of contact tracing apps during a pandemic?

Thank you for this comment. This term was mistakenly left and corrected to “new product growth model”. Apologies for this mistake.

Round 2

Reviewer 2 Report

The authors have addressed all comments in the first round of review. The authors have introduced a new section  “significance of the study” to cover the advantages and drawbacks in comparison with other similar proposals. This subsection is built and advances of the current work, not only from mHealth but also general technology adoption perspective is described in relation to related studies, adding 3 more references from indexed journals.

Furthermore, a new Figure 1 was added to describe a conceptual schema of the approach presented. Figure 1(a) describes the graphical representation of Bass’s product growth model and Figure 1(b) describes the conceptual schema of the approach employed in the study.

A new section was added to include a brief features description of the COVIDSafe App and COCOA app. In this new section, the reason for the selection of the apps (COVIDSafe: quick penetration, COCOA: slow and stepwise penetration) is described.

Finally, the Conclusion section was added, and potential future work is described in it.